# Recognition of Additive Manufacturing Parts Based on Neural Networks and Synthetic Training Data: A Generalized End-to-End Workflow

**Jonas Conrad** [1,*][iD], **Simon Rodriguez** [2], **Daniel Omidvarkarjan** [1], **Julian Ferchow** [1] **and Mirko Meboldt** [2]

1   inspire AG, Technoparkstrasse 1, 8005 Zürich, Switzerland
2   Product Development Group Zurich pd | z, ETH Zürich, Leonhardstrasse 21, 8092 Zürich, Switzerland
\*   Correspondence: jonas.conrad@inspire.ch; Tel.: +41-44-632-3602

**Abstract:** Additive manufacturing (AM) is becoming increasingly relevant among established manufacturing processes. AM parts must often be recognized to sort them for part- or order-specific post-processing. Typically, the part recognition is performed manually, which represents a bottleneck in the AM process chain. To address this challenge, a generalized end-to-end workflow for automated visual real-time recognition of AM parts is presented, optimized, and evaluated. In the workflow, synthetic training images are generated from digital AM part models via rendering. These images are used to train a neural network for image classification, which can recognize the printed AM parts without design adaptations. As each production batch can consist of new parts, the workflow is generalized to be applicable to individual batches without adaptation. Data generation, network training and image classification are optimized in terms of the hardware requirements and computational resources for industrial applicability at low cost. For this, the influences of the neural network structure, the integration of a physics simulation in the rendering process and the total number of training images per AM part are analyzed. The proposed workflow is evaluated in an industrial case study involving 215 distinct AM part geometries. Part classification accuracies of 99.04% (top three) and 90.37% (top one) are achieved.

**Keywords:** computer vision; deep learning; image classification; synthetic training data; additive manufacturing

## 1. Introduction

Additive manufacturing (AM), which is also referred to as 3D printing, has matured to become an established manufacturing process in various fields, e.g., prototyping or spare-part and end-user component production [1]. AM enables efficient low-volume production of highly mixed parts, e.g., customized industrial parts, including parts with complex geometries [2,3], because AM does not require special tools or molds and allows for fast lead times and decentralized production at small and medium-sized fabrication sites. However, frequently used powder-based AM processes, e.g., laser powder bed fusion [4] and Multi Jet Fusion (MJF) [5], often require extensive manual post-processing of nearly all parts [6,7]. Parts must be cleaned of excessive powder and packaged. Optionally, they can undergo surface treatments or dyeing processes. Note that such post-processing steps and the printing process can be realized in batches because multiple AM parts can be printed or post-processed simultaneously [8]. However, not all AM parts are processed in the same manner; thus, batches frequently require sorting between each processing step. An example is packaging, where AM parts are sorted before being packaged for shipping to different customers.

One critical subtask of part sorting is the recognition of parts. In [9], manual part recognition was identified as the primary time factor during the part sorting process, accounting for 40% of the total sorting time. Part recognition for sorting is particularly relevant for

non-serial production of AM parts due to frequent changes in part geometries, e.g., for prototyping and customized parts. In addition, there is more variance in part-specific post-processing than in serial production. Figure 1 illustrates the steps in a conventional AM process from printing to shipping as a series of batch processes. If the upcoming process step differs among the AM parts in a single batch, e.g., the parts will be dyed different colors, they require recognition.

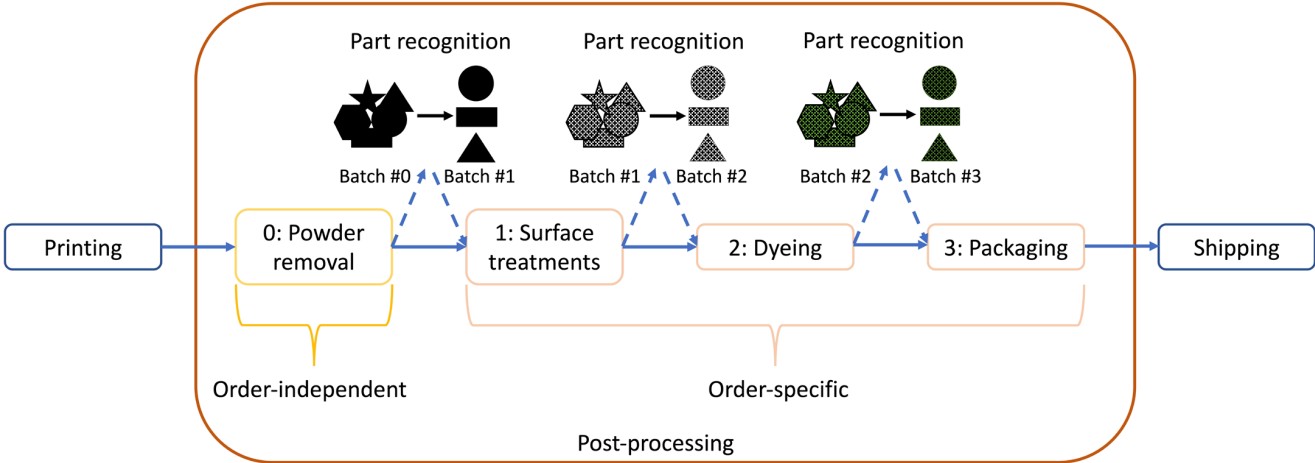

**Figure 1.** Part-specific post-processing of AM parts in batches that requires part recognition.

AM part recognition is commonly performed manually, which is time-consuming, prone to error and cost intensive. Workers recognize each part individually using part lists that contain images of the parts' 3D models [10]. The time and effort increase when the number of AM parts processed in one batch is continuously maximized to ensure efficiency, for example, by optimizing the use of the build volume [11] or increasing the dimensions of the build chambers. When more parts are processed simultaneously, more parts require recognition, which increases recognition times because the parts must be recognized from a larger selection of parts. Thus, with increasing production volumes, the low degree of automation in part recognition can lead to a major bottleneck in the overall AM production process. This further hinders full end-to-end automation of AM post-processing [12]. Therefore, manual recognition of batches of AM parts presents a key challenge in the current post-process chain of powder-based AM processes. Consequently, it would be desirable to automate the AM part recognition process to reduce the time and labor costs of AM part sorting. To benefit from its potential, AM must be realized as efficient distributed production, e.g., for spare parts. This should be realized with low-cost systems to ensure low initial investment costs for new AM production facilities [13].

One solution that simplifies AM part recognition and allows for automation is tagging, which was reviewed in [14]. Tags can be optical or chemical markers, or geometric features, e.g., QR codes or barcodes, or integrated electronics, e.g., RFID chips. While these solutions allow highly accurate differentiation of different parts, they require additional effort in the part design process. Here, each AM part must be adapted with an individualized tag, which is not feasible for high volumes of changing part geometries because the process cannot be automated for all part geometries. In addition, not all tagging solutions are suitable for all part geometries. Thus, a purely part geometry-based approach to part recognition could be beneficial. One approach that does not require tags is the use of deep learning and neural networks, which have been studied extensively [15,16]. A common application is image classification, where neural networks can be utilized to recognize the class of objects present in an image [17]. Thus, it is expected that neural networks can be used to recognize AM parts by classifying images of parts. Supervised learning techniques are frequently used to train a neural network for a given task, e.g., image classification. In supervised learning, labeled data, e.g., images, are required to adjust the network parameters (also

referred to as weights) to classify unlabeled data correctly [18]. Compared to conventional machine learning methods, such a learning process can be set up in an end-to-end structure, where only minimal human intervention is required [19]. In [20], it was demonstrated that neural networks can be employed to recognize AM parts. Here, different neural network structures for object detection were trained to classify and localize AM parts. The trained networks were then applied to recognize six different AM parts in images. However, this approach required capturing and labeling images of printed AM parts to train the network. As this process was performed manually, it is not feasible for the production of constantly changing AM geometries due to the high required effort. Acquiring labeled image data to train neural networks has been identified as a common challenge. Collecting and labeling training data can be time-consuming, and in some cases, it can be difficult to capture the required amount of training data [21]. Similar challenges, specifically for the application of neural networks in AM, were identified in [22,23]. The high effort and expertise required to create a training dataset and the lack of knowledge required to train a well-functioning neural network currently hinder application. To overcome these challenges in AM part recognition, the use of synthetic data to train neural networks for the classification of real images was mentioned in [20]. Here, training images can be generated as renderings from the AM parts' computer-aided design (CAD) files. Rather than capturing and labeling images, synthetic images and their corresponding labels can be created from digital 3D models. In addition, several previous studies have demonstrated that neural networks can be trained on synthetic images [24–26]. This method reduces the manual training costs significantly because it enables fully automated workflows to generate training data [27,28] without reducing the performance of the model [29]. CAD-based 3D models are a requirement in AM; thus, no additional costs are incurred to create them to generate training data. As a result, synthetic training data generation is suitable for AM processes. Previous studies have attempted to realize this approach based on synthetic training data to recognize AM parts. For example, neural networks for AM part recognition based on 2D data (images) and 3D data (point clouds and voxels) generated from CAD models were evaluated in [10]. Here, high part recognition accuracies were achieved; however, the authors did not address industrial application scenarios that require a generalized and optimized approach or specify the details required to reproduce their approach. In [9], a similar concept comprising a pipeline to train neural networks for image classification on rendered images was presented, which led to promising results in terms of recognizing AM parts with high accuracy. However, the authors did not describe all the necessary details, e.g., the required hardware or specific information about their evaluation methodology. A process based on synthetic training data used to classify and localize powder-covered AM parts was also developed in [12]. Here, a neural network was trained on renderings that included simulated powdering from AM part CAD models. With this method, the powdering was modeled using an artificial powder simulator and a physics simulation. During evaluation, high classification and localization accuracy was obtained. The evaluation was conducted using a test set comprising four distinct AM part geometries. The network structure evaluated by the authors was trained for 100 epochs, meaning that the entire training dataset was iterated 100 times during the training process, which is time-intensive. Brief training times are especially relevant in an industrial application where the part geometries vary in every production batch. Classification networks must be retrained for every batch with new part geometries in this scenario and training times must kept short to not exceed AM production requirements. Thus, methods with intensive training are infeasible for the recognition of AM geometries that change regularly.

In summary, previous studies have demonstrated the potential of using neural networks trained on renderings of AM parts' CAD models to recognize AM parts; however, the following limitations have been identified:

1. High-performance hardware is used for the recognition process, which is not applicable for most AM service providers due to high investment costs [13].

2. The computational resources required for data generation and network training are relatively high. This results in limited applicability in industrial settings where AM parts requiring recognition can vary daily, leading to training data generation and network training in high frequency. For this, data generation and training times must not exceed print times and be minimized for efficiency.

3. The processes required to generate training data and train neural networks and the required recognition hardware were not fully disclosed in the previous studies [9,10]. Thus, it is unclear if these previous methods are generalizable and would perform well in an industrial application with high variation in part geometries.

To overcome these limitations, in this study, an automated generalized workflow for visual AM part recognition using low-cost hardware is fully disclosed, optimized, and evaluated in an industrial case study. The focus is thereby set on further closing the gap for widespread industrial applicability in non-serial production, where each new post-processing batch potentially requires the training of a new network due to constantly changing part geometries. Training data generation and network training must therefore be generalized. This is opposed to serial production, where part geometries stay constant and only one recognition network could be trained and optimized for one specific set of parts.

An overview of the proposed workflow is shown in Figure 2. The proposed workflow is based on STL format CAD models from which images of parts are rendered. These synthetic images are used in a network training step to create an image classification network, which is then utilized in the final step to recognize AM parts automatically. In this paper, a detailed description of the proposed workflow for automated part recognition is provided. Here, the focus is industrial applicability with low-cost hardware; thus, the workflow is fully automated and generalized. In other words, no neural network training expertise is required to train networks for new AM part geometries. In addition, the proposed workflow is optimized in terms of computational resources using a specifically created test set of 30 distinct AM parts. This optimization allows for network training during the printing process and economic use of the workflow in distributed production sites. Real-time part recognition is made possible by deploying a neural network on a low-cost single-board computer. Finally, an industrial case study involving 215 distinct AM part geometries and 519 AM parts in total is conducted to evaluate the generalized workflow.

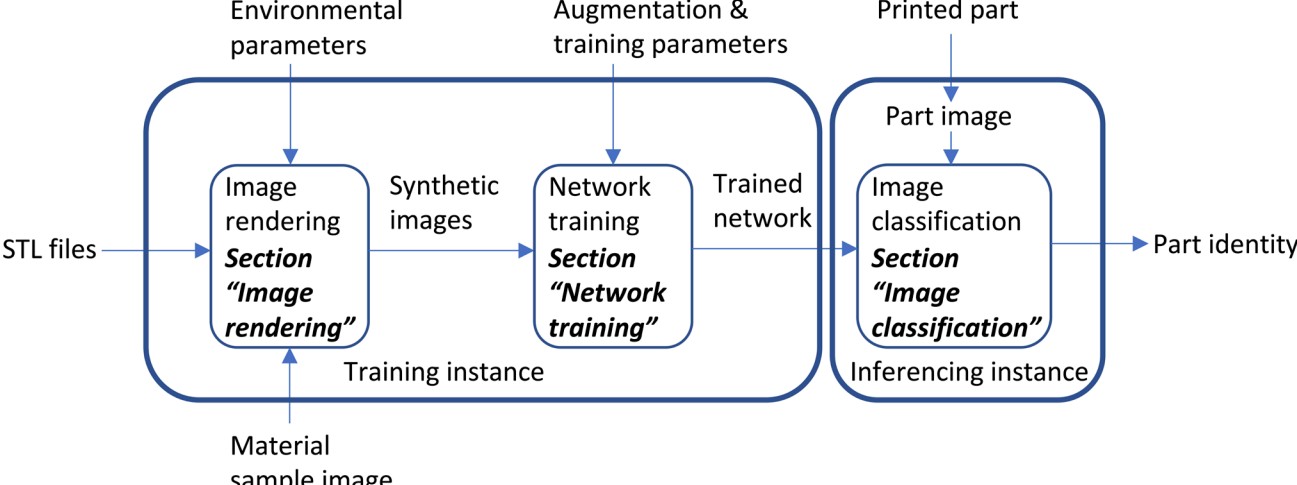

**Figure 2.** Overview of the proposed neural network-based end-to-end workflow for automatic visual recognition of AM parts.

Our primary contributions are summarized as follows:

- Contribution 1: A generalized neural network-based workflow for visual AM part recognition on low-cost hardware is proposed (Section 2);

- Contribution 2: The proposed workflow is optimized in terms of the required computational resources (Section 3);
- Contribution 3: The proposed workflow is evaluated in an industrial case study that solely includes previously unknown part geometries (Section 4).

All the contribution chapters contain the respective approach and corresponding results. This is followed by an overall discussion in Section 5. Finally, the paper is concluded in Section 6, including suggestions for potential future work.

## 2. Proposed Workflow

This section describes the proposed neural network-based end-to-end workflow (Contribution 1) in detail. The proposed workflow was developed to be executed automatically, with a directory including the AM parts' CAD files being the sole manual input. It was generalized to be applicable to a wide range of AM part geometries without adaptations. Figure 2 shows an overview of the proposed workflow, which includes three sub-steps: image rendering (Section 2.1), network training (Section 2.2) and image classification (Section 2.3). These three steps are described in the following subsections.

### 2.1. Image Rendering

The automated workflow begins with the image rendering step, which generates the synthetic data for the subsequent network training step. Here, the synthetic training data comprise images generated from CAD models of the AM parts to be recognized. To generate the images, each CAD model is processed in the same manner using Blender, which is a free computer graphics tool. First, the CAD model is loaded from an STL file as part of a rendering scene. Then, the model's surface appearance is recreated using a material source image. Note that this material image must be captured manually once for each printed material. Afterwards, a noise texture is added to simulate possible irregularities in the appearance of the surface. Such irregularities are frequently caused by the fabrication process itself or by post-processing steps, e.g., sandblasting.

After the part is loaded into the rendering scene, it is positioned on the ground plate by simulating drops from an elevated position. As a result, the initial orientation of the part is varied by fixed the degrees around the two axes making up an orthogonal triple with the direction of gravity. This is done to introduce variation into the positioning of the part. Then, the influence of gravity on the part is simulated to make it fall onto the ground plate. These falls are simulated to include viable part orientations on the ground plate under the influence of gravity, which allows for more realistic shadow rendering [26]. Once the part is positioned, images from different simulated camera angles are rendered using the Cycles rendering engine. Here, the cameras are equally distanced on a circle above the ground plate and angled toward the part. The images are rendered at a resolution of $256 \times 256$ pixels and saved in JPEG format. The resolution differs from the final network input ($224 \times 224$ pixels) to allow for randomly cropping the image for enhanced variation in the training data. The following parameters are defined as the final workflow characteristics:

- Physics simulation (gravity): On;
- Number of simulated drops per part: 64;
- Number of camera perspectives per part: 4;
- Total number of training images per part: 256.

The influence of including simulated gravity in the rendering process and the number of camera perspectives on the classification accuracy is analyzed in Sections 3.2.2 and 3.2.3.

Figure 3 shows the rendering scene and example synthetic images. The scene was designed to resemble the actual image classification scenery, which is referred to as the sensing area. It comprises a white ground plate and nine light sources. Here, one main light source (2.5 Watts) simulates a light-emitting diode (LED) panel that is mounted above the image sensing area, and eight secondary light sources (0.4 Watts) simulate the influence of atmospheric light and its reflection. These lights are positioned at equal distances in a circle above the ground plate. In addition, a Cartesian coordinate system is used to define

positions in the rendering scene. Its origin is placed in the geometric center of the ground plate, and the *z*-axis lies parallel to the ground plate's normal vector.

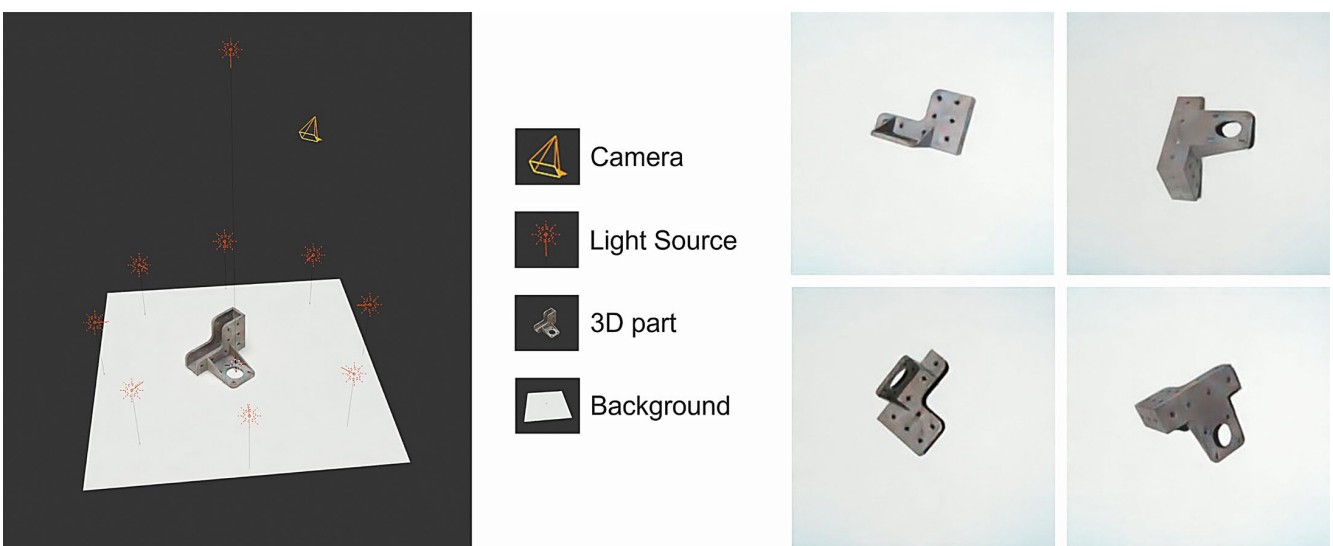

**Figure 3.** Image rendering scene and examples of synthetic images of an AM part at different orientations.

The entire rendering process is realized on a training instance using a Nvidia QUADRO RTX 4000 graphics processing unit (Nvidia Corporation, Santa Clara, CA, USA). Note that the computational resources required for the training instance are considerably greater than those for the inferencing instance. Thus, the training instance should be connected to multiple inferencing instances for network training to realize low system costs.

## 2.2. Network Training

In the network training step, a neural network for image classification (VGG16 [30]) is trained on the previously generated synthetic images. Here, the training procedure is a supervised learning process (Figure 4). First, a network is initialized and a prediction is made using the network. Based on the prediction, the current training step's loss and gradient are calculated to adjust the weights of the neural network. This process is repeated until a stopping criterion is reached. Multiple network structures for image classification were evaluated before deciding on VGG16. This was intended to find a network structure that allows for fast inference and high classification accuracy for AM parts when trained on synthetic images. The influence of the network structure on the part classification accuracy is described in Section 3.2.1.

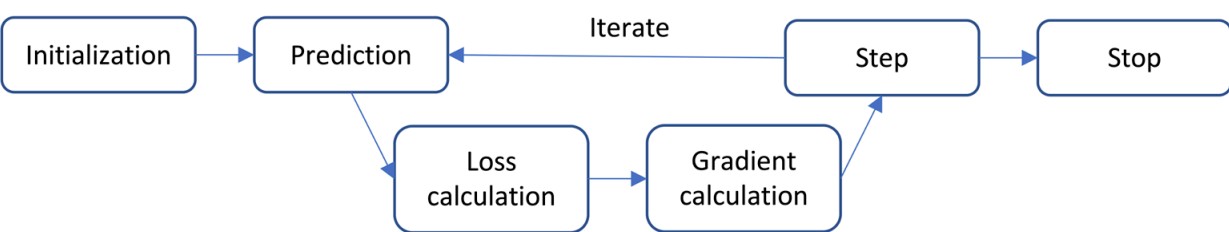

**Figure 4.** Supervised learning process used to train the neural network.

For the training process, the synthetic training data are randomly split into 80% (training set) and 20% (validation set). The split excludes a test set, as during industrial application, there exists no labeled test data. All the testing for development and evaluation was realized using a separate dataset containing only real images (Section 3.1). During industrial application, each batch containing new parts to be recognized requires the training of a new network. This is necessary, as in the application scenario, AM parts that

are to be produced are not known until ordered by a customer and the geometries can vary strongly. Testing prior to application is not possible without manually generating a test set, which would contradict the workflow's purpose. The training is realized on the training instance using the following parameters:

- Input image size: $224 \times 224$;
- Batch size: 32;
- Learning rate: $5 \times 10^{-5}$;
- Number of training epochs: 5;
- Loss function: categorical cross-entropy loss.

These parameters have been defined with the help of the test set (Section 3.1). They were varied to reach a low number of network weights, short training and inference times, and high classification accuracy. Especially relevant to this is the input image size. Not only does a greater model input size increase training and inference times, it also leads to an increase in the rendering resolution for the synthetic images. Finally, this increases the duration of the rendering process. The input image size was therefore chosen to be as small as possible while still maintaining high part classification accuracy.

The network is pretrained on a dataset of general objects (COCO, [31]), which reduces the training effort required to classify specific AM parts. This is an established practice known as transfer learning [32]. In addition, data augmentation is utilized to prevent overfitting, where the trained network too closely matches the training data. When overfitting occurs, the network does not generalize well to previously unseen data (i.e., the test data) [33]. The images are augmented using the following factors:

- Zoom range: [0.95, 1.05];
- Brightness range: [0.8, 1.15];
- Rotation range: [−45, 45];
- Height shift range: [−25, 25];
- Width shift range: [−25, 25].

The network is compressed in the network training step to allow for inference on the low-cost inference instance. Here, the compression is realized by reducing the network's weights' precision by switching their format from a 32-bit float to an 8-bit float. This is known as quantization [34], and including this limitation in the training process is known as quantization-aware training. Typically, quantization-aware training leads to higher network accuracy than realizing quantization after the training process [35]. Before transferring the network to the inferencing instance, it is converted to the TensorFlow Lite format and compiled for hardware compatibility.

The combined process time for the image rendering and network training was measured as 8 min 35 s per part for a test set containing 30 parts, resulting in a total duration of 4 h, 17 min, and 45 s. Note that this is considerably less than the printing and cooldown time for a typical MJF print job (approximately 16 h, depending on the machine settings). Furthermore, the process time depends on the performance of the selected training instance.

*2.3. Image Classification*

In the final step of the proposed workflow, part images are acquired and classified using the previously trained neural network. Based on the classification results, the parts are recognized, and the result is communicated to the user. The part recognition is implemented using a physical low-cost setup, which is shown in Figure 5. It comprises multiple laser-cut parts encasing a ground plate, which is referred to as the sensing area (37 cm × 40 cm). The sensing area's background is adaptable to ensure maximum contrast for the AM parts placed on the backplate. This configuration also integrates an LED panel to realize consistent lighting of the sensing area, a camera (Arducam Autofocus, 5 Megapixel (Arducam Technology Co., Hong Kong, China) to capture images of the parts placed in the sensing area and the inferencing instance (Raspberry Pi 4 8 GB model B (Raspberry Pi Foundation, Cambridge, UK) with a Coral Edge Tensor Processing Unit (TPU, Google

LLC, Mountain View, CA, USA)), which applies the trained neural network. Note that the components, excluding a display device, can be acquired for approximately USD 200. The required resolution and accuracy of the camera and the computational resources of the Raspberry Pi and TPU are comparatively low and scale with the price. As these are the main technical components, the setup could be integrated into existing machines or unpacking stations, as they would only need to be extended by a camera, a Raspberry Pi or comparable computing unit and a lighting unit. Some manufacturers already include these components in their machines.

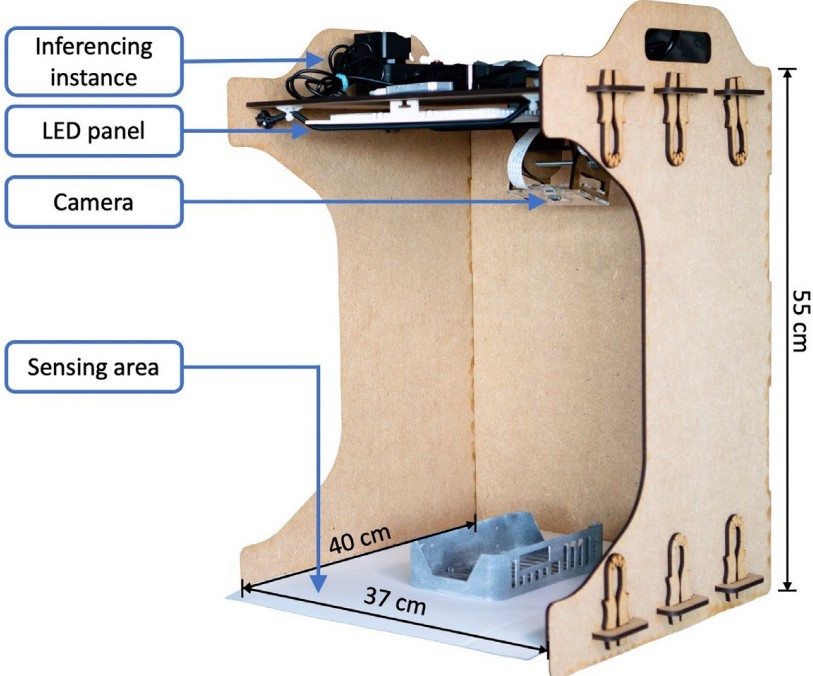

**Figure 5.** Low-cost setup used for the automated recognition of AM parts, comprising the sensing area, a camera to capture images of the part in the sensing area, an LED panel for consistent lighting conditions and an inferencing instance running the image classification network.

As shown in Figure 6, a graphical user interface (GUI) can be used to select a previously transferred image classification network on the inferencing instance and begin the part recognition process. Here, the AM parts to be recognized are placed manually in the sensing area. Then, the top-three class predictions for the given part are displayed in real time on the GUI with a refresh rate of 3–4 frames per second. The classification results are accompanied by a live view of the sensing area and images of the top three predicted parts. The user can evaluate the part information from the GUI by comparing the actual part to the classification results. Thus, the final recognition step is performed by the user; however, the selection is limited to only three different part geometries, drastically accelerating the recognition process. This allows for more reliable part recognition without the requirement of 100% top-one classification accuracy, which would be very difficult to achieve because printable part geometries are very diverse. Ensuring 100% accuracy by testing all possible part geometries would be highly infeasible in this application scenario.

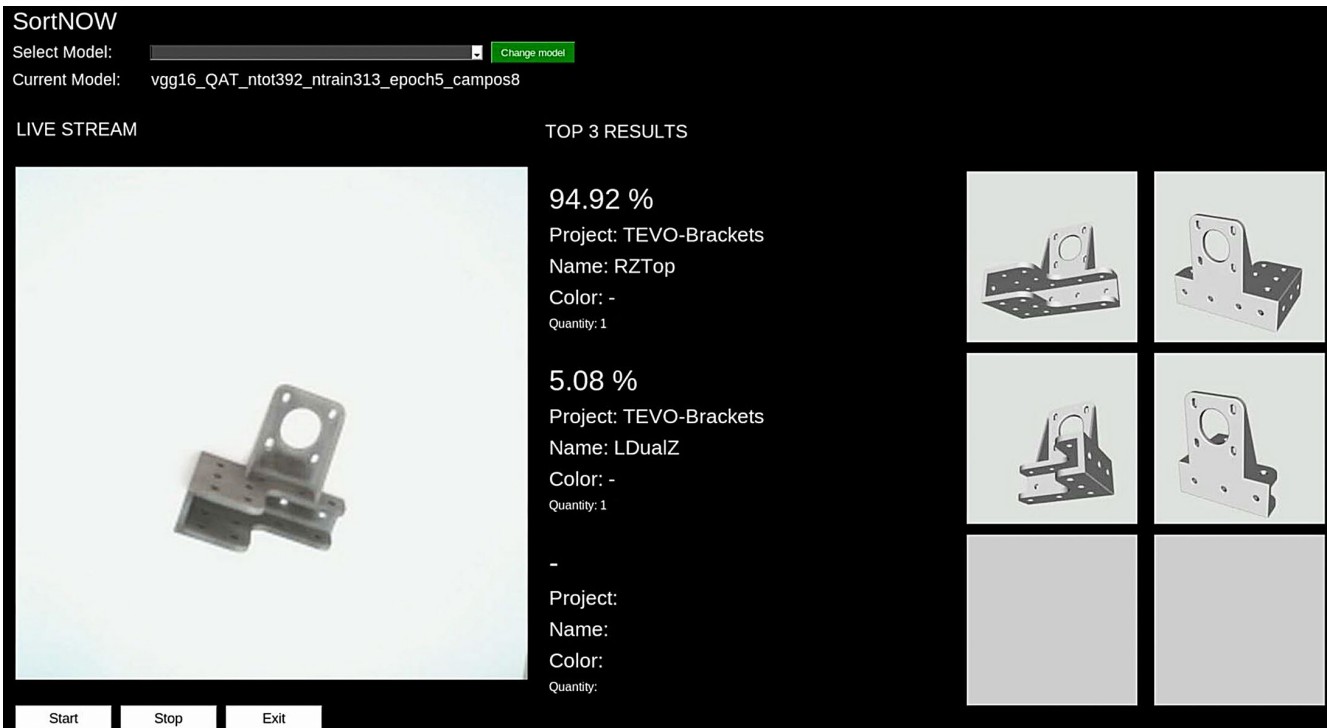

**Figure 6.** GUI used to display the recognized AM part after recognition, including a live view of the sensing area (**left**), the top three proposed part identities (**middle**) and renderings of the top three propositions (**right**).

## 3. Optimization

For Contribution 2, an in-depth analysis of multiple workflow characteristics was performed. The analyzed characteristics are closely related to the required computational resources and, therefore, influence the duration of the process. Here, the goal was to limit the process duration while targeting high part classification accuracy. For this, a test set was developed specifically.

### 3.1. Test Set

A test set containing solely real part images was collected for development and evaluation purposes. The test set comprised 30 distinct AM parts. The corresponding CAD models were collected from eight different projects from thingiverse.com, which is a platform for open-source hardware designs. Note that the AM parts used in this evaluation can be found via the corresponding project number in the references [36]. The selection included parts with footprints ranging from 3 cm × 3 cm to 20 cm × 11.5 cm. Parts were selected with the goal of including various part geometries and sizes and allowing for the development and optimization of a generalized workflow, which can be used to train further networks for the recognition of new AM parts. Figure 7 shows the digital models of all the test parts. In this study, all the parts were produced from PA 12 (Nylon 12) via MJF. After printing, all the parts were randomly placed in the sensing area 40 times to record a total of 1200 test images. The test set is available under the following link: https://data.mendeley.com/datasets/trd8nry345/1 (accessed on 23 October 2023).

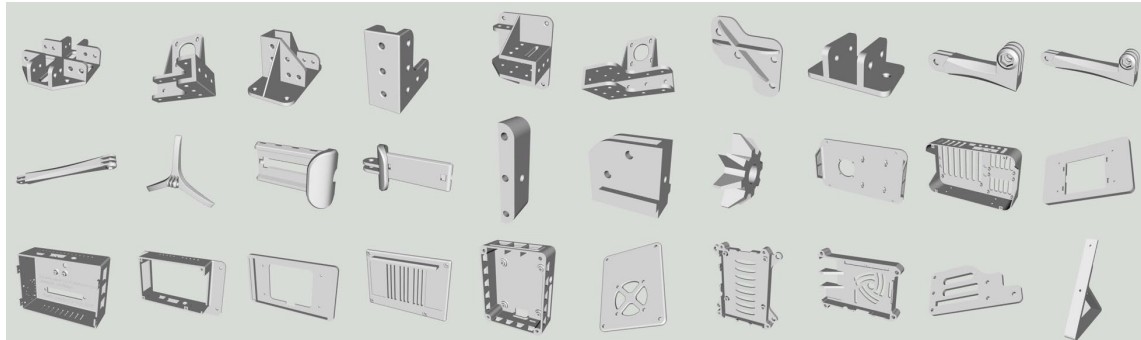

**Figure 7.** Renderings of test parts for the evaluation of the proposed workflow.

### 3.2. Optimized Workflow Characteristics

The test set was used to analyze the influence of several workflow characteristics on the AM part classification accuracy, i.e., the selected neural network structure, the integration of a physics simulation, the number of camera positions and the number of rendered images per part. All these characteristics influence the workflow in terms of the required computational resources and recognition accuracy and can therefore be adapted to optimize the workflow. Their specific influence is described in the corresponding subsections.

In the upcoming figures, the part classification accuracy is presented in box plots to visualize the variance in the accuracy over the 30 different part geometries included in the test set. Each part geometry was classified 40 times. The box plots display the following information on the accuracy distribution over all the test part classes: median and lower and upper quartile (box), the interquartile range (IQR) between the 0.25 quantile and 0.75 quantile (whiskers), and outliers, which are defined as values that are 1.5 times IQR bigger or smaller than the upper or lower quartile (x-marks).

#### 3.2.1. Neural Network Structure

The structure of the neural network influences the classification accuracy. In this evaluation, three common network structures were considered, and they were selected according to their compatibility and computational efficiency with the low-cost hardware. The following network structures were evaluated:

- MobileNetV2 [37] (trained for five and twenty epochs)
- ResNet-50 [38] (trained for five epochs)
- VGG16 [30] (trained for five epochs)

MobilenNetV2 is built on a very simple network structure made up of 53 layers, focusing on mobile applications. It uses comparatively few parameters (3.4 million) to achieve good classification performance at low computational costs [37]. The inference is about 50% faster than with RestNet-50 or VGG16.

RestNet-50 consists of 50 layers with 25.6 million parameters. With the introduction of residual learning, network optimization for a specific application became easier, even for networks with increased numbers of layers [38].

VGG16 is the biggest of all the tested networks, with a total of 138.4 million parameters in 16 convolutional layers [30]. However, when deploying it on the proposed hardware, including the TPU, it classifies about 3–4 frames per second. All the network structures are presented in detail in the corresponding articles.

All the networks were trained for five epochs on the test set for a varying number of training images per part. MobileNetV2 was also trained for 20 epochs because it was found that training for 5 epochs achieved relatively low part classification accuracy.

Figure 8 shows the influence of the neural network structure on the classification accuracy for different numbers of training images per part. Here, the accuracy distribution of MobileNetV2 is shown in blue (five epochs) and red (twenty epochs), ResNet-50 is shown in yellow, and VGG-16 is shown in purple.

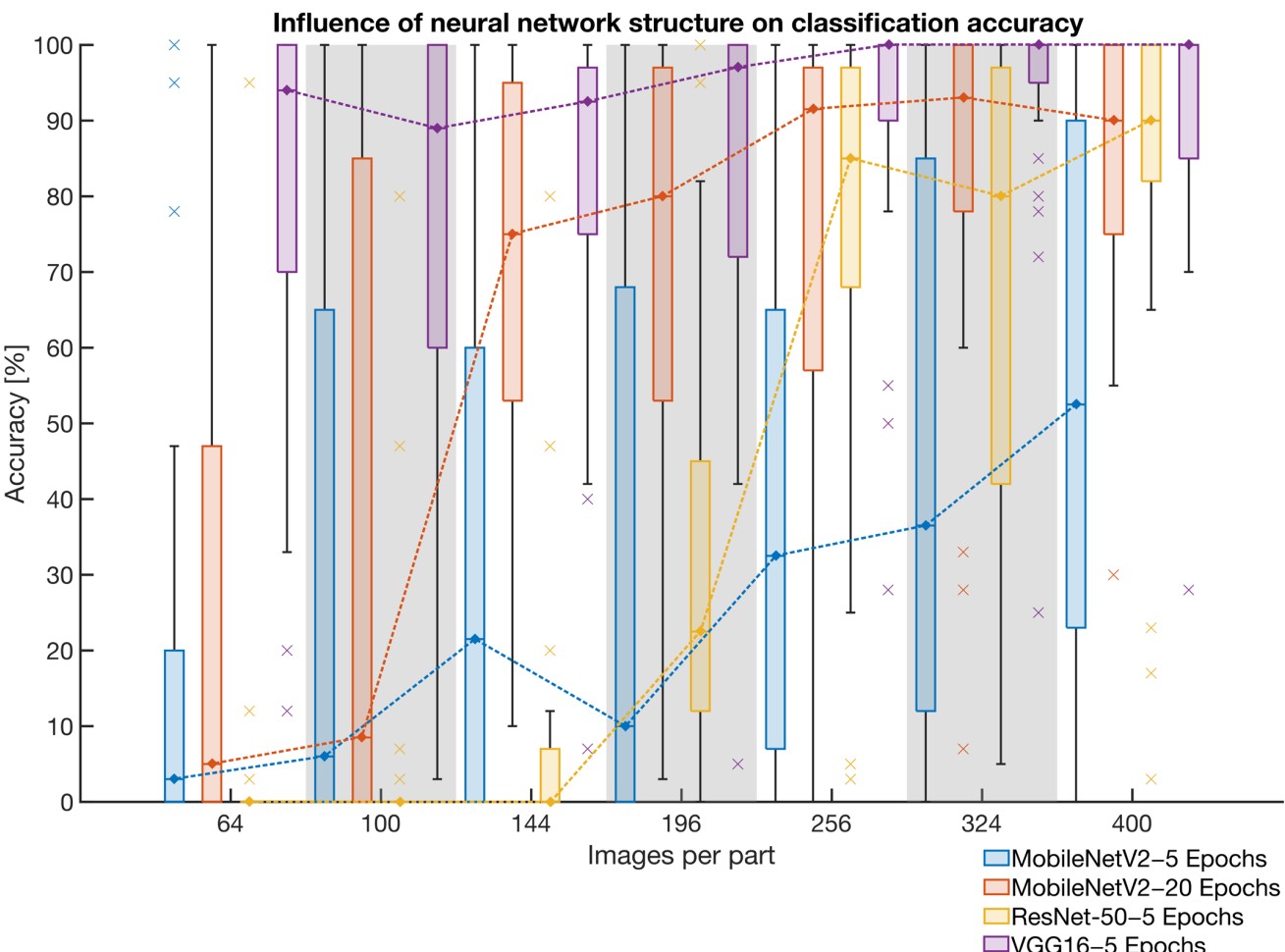

**Figure 8.** Comparison of the different network structures' classification accuracy on the test set. The networks were trained on different numbers of images for either five or twenty epochs.

As can be seen, the VGG-16 network achieved the highest average part classification accuracy of 93.17% on the test set. The influence of the network structure was particularly evident at low numbers of training images per part, where ResNet-50 reached average accuracies that were close to 0%, and the lowest accuracy achieved by the VGG16 network was 76.20%. With high numbers of training images, the difference between the network structures was reduced but was still evident. As in previous experiments, VGG-16 reached a maximum accuracy before dropping again with 400 training images per part. It was found that increasing the number of training epochs appeared to benefit MobileNetV2. However, this also resulted in a 400% increase in the network training effort, and the maximum accuracy was still less than that obtained by the VGG-16 network. Considering these results, VGG-16 was selected as the network structure for the proposed workflow.

### 3.2.2. Physics Simulation

A physics simulation was considered to increase the shadow rendering quality and include only viable part orientations in order to reduce the number of training images required to reach the maximum classification accuracy. Here, images of unlikely part orientations would be excluded by simulating the effect of gravity when dropping the parts onto a plane, mimicking a worker placing the parts. This is illustrated in Figure 9. The probability that a worker would place the pictured part on its smaller face is considerably lower compared to it being placed on one of its greater faces, as he would seek to balance it, which is unlikely in an industrial scenario. Note that fewer training images would result in lower rendering costs during the proposed workflow. As the computational effort

required for the physics simulation is neglectable compared to the image rendering process that includes realistic lighting and shadow casting, an integration could be beneficial to accelerate the workflow.

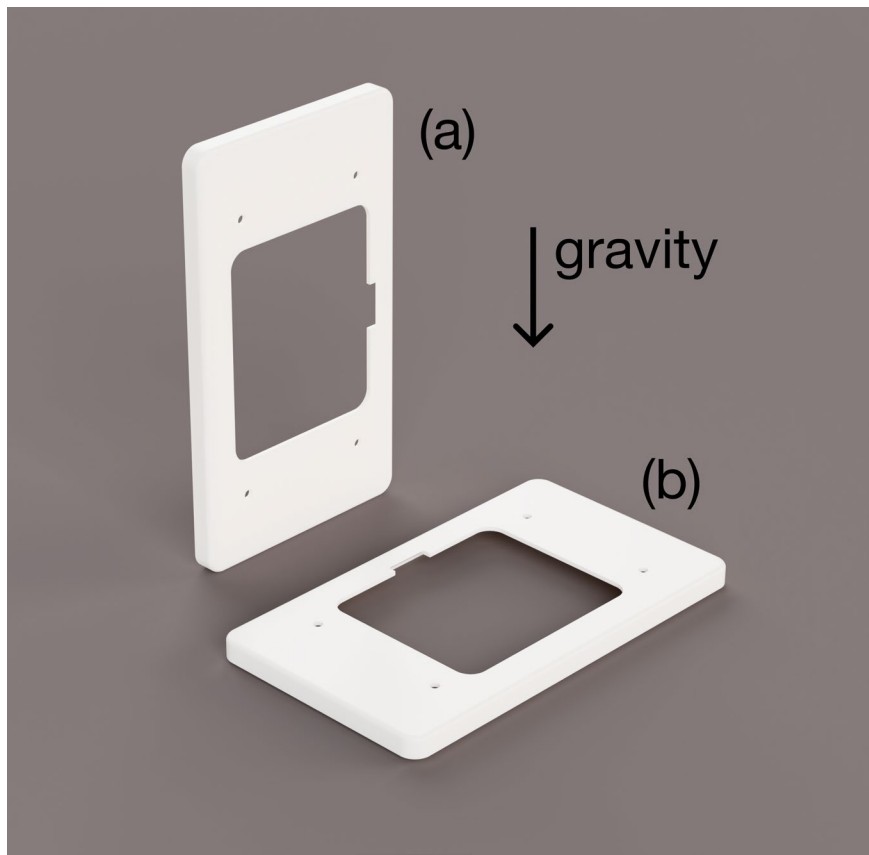

**Figure 9.** The same part placed in an improbable orientation (**a**) standing up on a small face and in a probable orientation (**b**) laying down on a greater face on a ground plate, under the consideration of gravity.

The influence of including simulated gravity was evaluated by classifying the developed test set using the VGG-16 network trained on different numbers of training images with and without the physics simulation. In this evaluation, the parts were placed in varying orientations directly on the ground plate for the networks trained without the gravity simulation.

Figure 10 shows the influence of including the gravity simulation on the classification accuracy on the test set for different numbers of images per part. Here, the classification accuracy distribution obtained with the gravity simulation is shown in red, and that obtained without the gravity simulation is shown in blue. As can be seen, the accuracy increased with an increasing number of images for both cases before slightly dropping again at 400 images. On average, the accuracy obtained with the gravity simulation was 5.38% higher. It further showed lower variation. Based on these findings, it was decided to include the physics simulation in the proposed workflow.

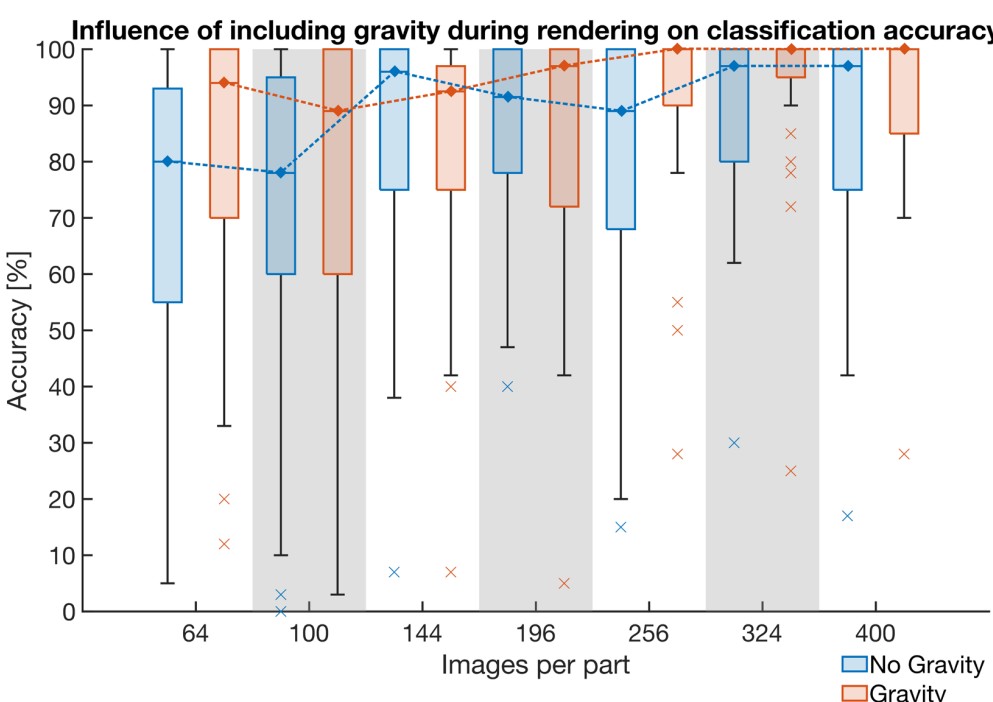

**Figure 10.** Influence of the gravity simulation in the proposed workflow on the accuracy when classifying the test set.

### 3.2.3. Number of Camera Positions and Number of Training Images per Part

The number of training images per part influences the part recognition accuracy. In this evaluation, a VGG-16 network was trained on different numbers of images per part to identify the optimal number images that must be rendered to achieve high classification accuracy on the test set. As the number of images per part is directly related to the number of simulated camera positions, the influence of using one, four and eight camera positions was also evaluated. Here, it was found that reducing the number of training images per part reduced the computational costs of the rendering.

Figure 11 shows the influence of different numbers of simulated camera positions on the classification accuracy with various numbers of training images per part. Here, the accuracy distributions for one, four and eight camera positions are shown in blue, yellow, and red, respectively. It was found that the accuracy obtained with eight camera positions increased continuously after exceeding 128 training images per part. In contrast, the classification accuracy was reduced with 400 images for one and four camera positions. Using multiple camera positions resulted in higher accuracy compared to using only one camera. For the proposed workflow, the number of training images per part was set to 256 with four camera positions because the increase in accuracy for 288, 324 and 392 images was low and increased the rendering costs. We observed that the maximum average accuracy increases by 6.18% for a 53.12% increase in the number of training images.

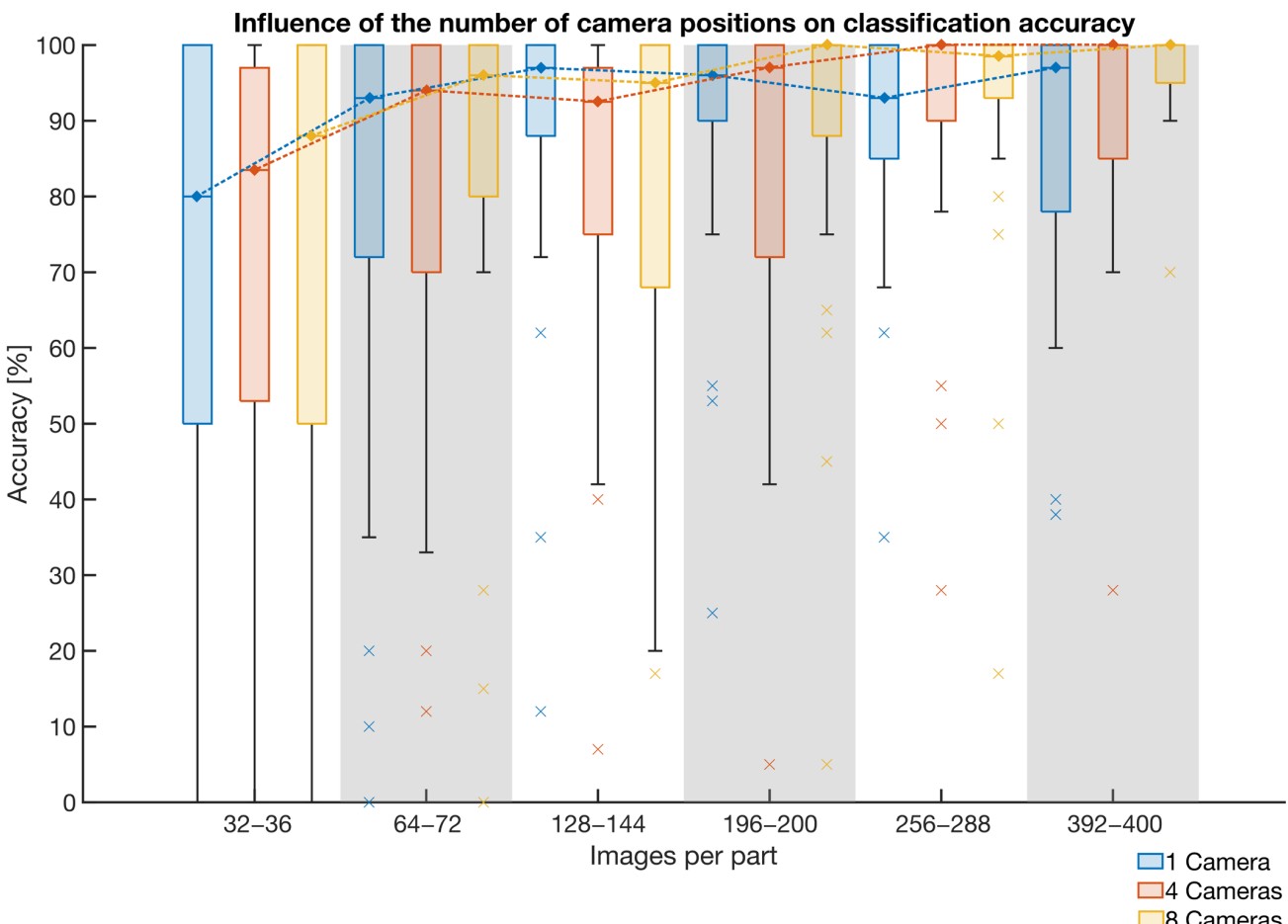

**Figure 11.** Comparison of the set classification accuracy obtained on the test set with different numbers of camera positions in the rendering scene.

## 4. Evaluation of Industrial Applicability

To support Contribution 3, the potential industrial applicability of the generalized workflow was evaluated in an industrial case study. Here, the proposed workflow was used to recognize previously unknown AM parts in the live production of an AM service provider.

### 4.1. Case Study

The case study was realized in partnership with an institution that offers AM part production using printers for MJF. Here, an HP Jet Fusion 4200 printer was used to fabricate parts from PA 12 (Nylon 12). In addition, the part post-processing included powder removal, sandblasting, optional dyeing, and sorting of the AM parts for customer pickup. The facility provides MJF production as a service to universities. Two to three build jobs are produced per week, and the facility can produce approximately 60–100 parts each week. The part geometries generally differ between build jobs.

For this case study, the proposed workflow was set up and the parts were recognized automatically using the sorting system (Figure 12). Six different build jobs were analyzed, including a total of 519 printed AM parts with 215 distinct geometries, meaning that six different classification networks were trained. The system's ability to accurately recognize parts was analyzed. For this, the top-one and top-three part classification accuracies were measured. Accuracy is defined as the fraction of correct part classifications by the neural network. For top-one accuracy, the highest ranked network prediction must match the true part class, while for top-three accuracy, the three highest ranked network predictions must

match the true class. As the proposed GUI presents the three highest network predictions (Figure 6), top-three accuracy was also analyzed.

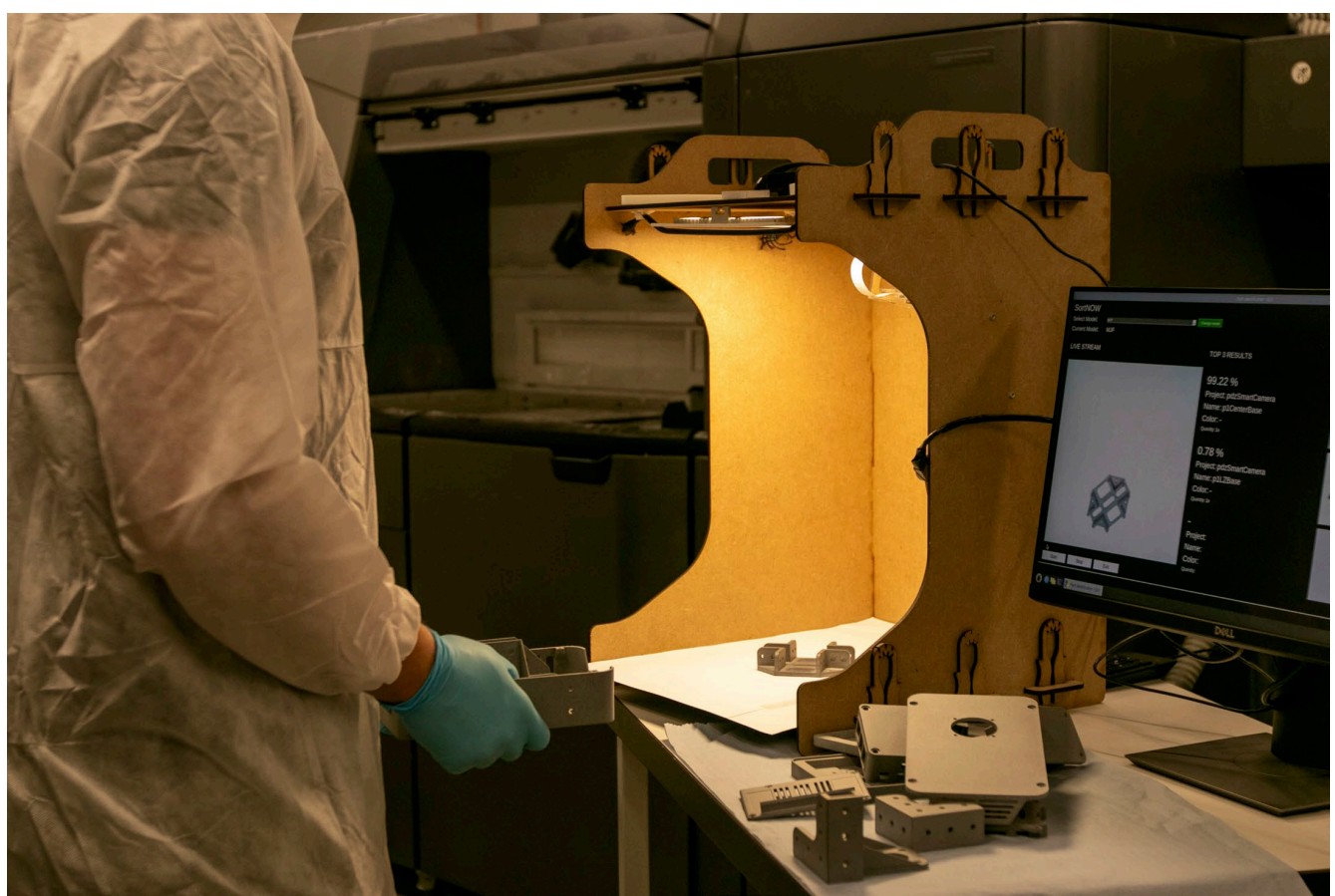

**Figure 12.** Proposed system for the automated recognition of AM parts during the case study. A worker is placing parts in the sensing area (left) and receives feedback about each recognized part via the GUI (right) in order to sort them.

To acquire information about the impact of the automated part recognition process on the sorting times, two build jobs were sorted once manually and once using automated part recognition, and the total sorting times were measured. The two build jobs involved 100 parts and 63 distinct geometries.

### 4.2. Case Study Results

Figure 13 shows the average part classification accuracy obtained in the case study. Here, each data group represents a single build job split into the top-one (blue) and top-three (red) accuracies. The overall top-one and top-three classification accuracies, including all the build jobs, are represented by the vertical lines. It was found that average part recognition accuracies of 99.04% (top three) and 90.37% (top one) were achieved for all the build jobs combined. In addition, little variation in the average accuracy was observed between build jobs, i.e., the standard deviation was 3.78% for the top-one accuracy and 1.20% for the top-three accuracy. Note that 100% top-one accuracy was not achieved. However, 100% top-three accuracy was obtained for three different build jobs.

Misclassifications were primarily observed with mirrored parts, parts that were too large to fit in the sensing area (37 cm $\times$ 40 cm), comparatively small parts and parts that only differed in scale from other parts. Here, small parts were defined as parts with a footprint smaller than 2 cm $\times$ 2 cm. In total, 5 out of 519 parts could not be recognized by the proposed workflow.

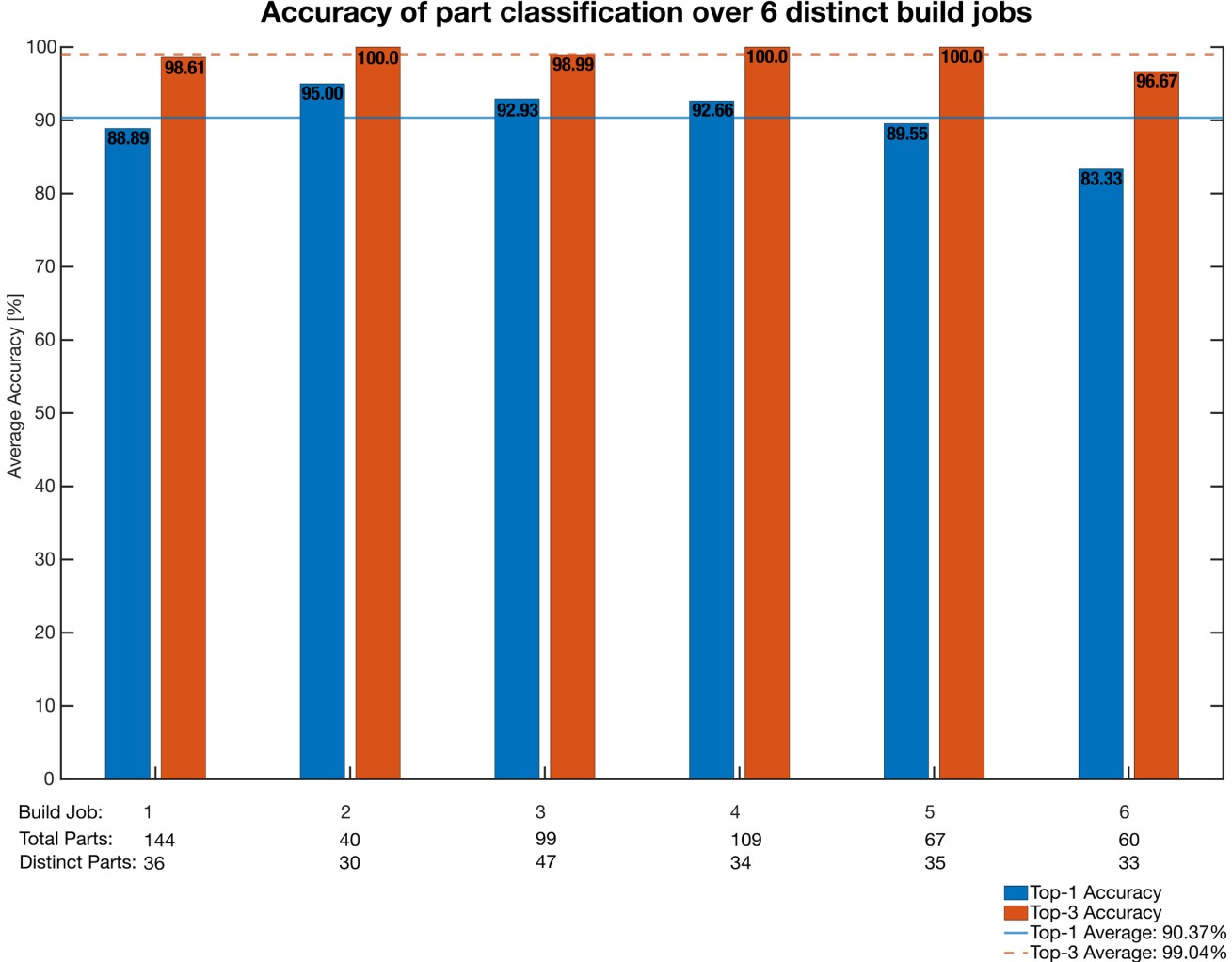

**Figure 13.** Average classification accuracy (top one and top three) for different build jobs.

The sorting times were 24 min and 50 s for manual sorting of 100 parts and 7 min and 47 s for sorting with the automated part recognition process. Thus, the duration of the manual sorting process was approximately 3.19 times longer than that of the automated sorting process.

## 5. Discussion

An overall discussion of Contributions 1–3 and the identified limitations is presented in the following subsections.

### 5.1. Contribution 1

Regarding Contribution 1, it has been effectively demonstrated that neural networks can be trained based solely on images created from CAD data for the task of visually recognizing real AM parts with low-cost hardware. In addition, this part recognition approach can be realized without design adaptations for tagging, as discussed in [14].

Furthermore, the proposed workflow addresses the challenges associated with the application of neural networks in AM, as highlighted in the literature [22,23], including the need for domain expertise and significant efforts for data creation. The networks used to recognize previously unseen AM parts can be created automatically, and only minimal manual input is required. The inferencing system has a comparably low cost and network creation times are shorter than the production times, which is a requirement for

industrial applicability, especially for distributed production with AM, where a high degree of automation and low initial investment costs are mandatory [13].

Note that the high costs of part sorting for batch processes are not exclusive to AM; thus, the proposed workflow could potentially be applied in other production scenarios where batches include a mixture of part geometries. One example is the sorting of waste machine parts after dismantling for recycling [39]. Here, the only requirement to automate part recognition would be the acquisition of digital models of the relevant parts.

*5.2. Contribution 2*

To support Contribution 2, the influence of the workflow characteristics was evaluated using a specifically created test set. Here, it has been found that the choice of neural network structure, the inclusion or exclusion of a physics simulation, the number of camera positions and the number of training images per part all influenced the AM part classification accuracy on the test set. Similar articles did not provide similar insights into workflow characteristics and their influence on AM part recognition [9,10]. The acquired knowledge was then used to optimize the proposed workflow to obtain high accuracy while maintaining sufficiently low computational costs. Currently, the results suggest that the VGG-16 network structure is best suited for image classification with the proposed workflow. This network structure achieved relatively high classification accuracy even for low numbers of training images compared to the MobileNetV2 and ResNet-50 networks. This is interesting, as all the network structures show comparable classification performance on a standard evaluation dataset [30,37,38]. In comparison, VGG-16 has the most network parameters by far (138.4 million), which seems to be beneficial for the proposed application. The high number of network parameters also causes the longest inference time of all the compared networks, although it still allows for 3–4 part classifications per second. In addition, including a physics simulation in the proposed workflow to only render AM part images in realistic orientations and training the network with 256 images per part from 4 camera positions for 5 epochs was found to optimize the workflow in terms of the computational resources and classification accuracy. In [12], 1000 synthetic images were used in total for 4 distinct parts (training and validation) and the training lasted 100 epochs, as comparison. As the image rendering and neural network training processes are computationally expensive, it is important to minimize the number of training images and training epochs without compromising the classification accuracy. This is particularly relevant when recognizing AM parts in non-serial production. In this scenario, parts can vary daily, and new classification networks must be trained for each new production batch. Since information on the analyzed workflow characteristics or the reasoning behind which characteristics were chosen was missing from previously published studies, the current work allows for easier recreation and optimization of automated visual AM part recognition and further research.

*5.3. Contribution 3*

The results of the industrial case study of the generalized workflow suggest that automatic AM part recognition can be realized using neural networks. The achieved part classification accuracies of 99.04% (top three) and 90.37% (top one) are comparable to the results of similar studies, which achieved part classification accuracies of around 93% (top one) [10] and 80–95% (top one) [9] with partially undisclosed methods and test data. In comparison to these studies, the underlying article discloses all the hardware and software components of the proposed workflow. Furthermore, all the tested parts were documented and allowed for specific findings on challenging part geometries. In general, the results suggest that the automated workflow using synthetic training data can be applied in industrial scenarios that include previously unknown parts, which, to the best of our knowledge, has not yet been investigated extensively and reproduceable in the field of automated post-processing for AM. The high part classification accuracies for all the build jobs substantiate that the workflow can be transferred from the test set used for

development to application with constantly changing part geometries. Furthermore, the proposed workflow appears to scale well to increasing numbers of parts in a production batch because the classification accuracy did not seem to be affected by the number of parts in the classification network. This is highly relevant because manual part sorting times increase nonlinearly with increasing numbers of AM parts in a batch that requires sorting. The study results additionally suggest that sorting times can be reduced significantly by using the proposed approach to automate part recognition. This was expected, as part recognition has been identified as the main contributor to manual part sorting times [9].

*5.4. Limitations*

Overall, the achieved results are promising regarding the industrial application of the proposed workflow; however, they also highlight certain limitations. Currently, full automation of the sorting process, e.g., using a picking robot, cannot be achieved because this would require 100% classification accuracy. However, obtaining error-free part classification for an unlimited number of part geometries is likely impossible. To date, including a final manual step to recognize the correct part from a selection of three parts seems beneficial because this technique is still considerably faster than recognizing a single part in a batch of many. In our case study, the maximum number of parts per build job was 144, from which the printed parts had to be recognized. As the reduction in sorting time could show, the obtained top-three accuracy of 99.04% is sufficient for practical application. However, this could be improved further by addressing the classification of critical part categories, e.g., particularly large or small parts, mirrored parts or parts that only differ in scale. A first experiment was conducted to gain further information on using neural networks to distinguish scaled or mirrored parts. It included four AM parts that were scaled and mirrored versions of the same part geometry. With enhanced training times and additional training images (20 epochs, 400 synthetic images), the part classification accuracy was observed to increase. This suggests a starting point for further research, as the findings would need to be supported by a bigger test dataset. Other potential approaches are the integration of further information in the classification process, e.g., part weight, to better distinguish scaled parts, or reinitializing the deployed networks weights stemming from the pre-training on the COCO dataset on another dataset that includes mirrored objects in specific.

In addition, the generalizability of our findings is limited because the test set only included 30 distinct AM part geometries, and the industrial case study only considered 519 AM parts and 215 distinct AM part geometries. Thus, our findings can only provide an estimate of the classification accuracy that could be obtained with an unlimited variation in part geometries. However, this estimate can be considered valid because the test set was created to resemble a typical AM production order, including parts of various sizes and geometries. Furthermore, the industrial case study included several build jobs from real-world customers and covered a wide range of part geometries. In addition, only MJF parts with changing outer geometries were included in the evaluation; however, the influence of the selected AM process on the classification accuracy is expected to be low. Note that an evaluation of the proposed workflow on additional AM parts and processes, e.g., selective laser sintering, is planned. For the recognition of parts that only differ in inner geometries, additional sensor systems are tested.

## 6. Conclusions

In this paper, a fully automated and generalized end-to-end workflow for visual recognition of AM parts that deploys on low-cost hardware was presented, optimized, and evaluated in a case study. First, the workflow sub-steps of image rendering, network training, and image classification were described in detail. The real-time image classification was realized with hardware components for approximately USD 200, and the image rendering and training to recognize a new batch of AM parts took four to five hours. Second, the workflow characteristics of the deployed neural network structure, the integration of a

physics simulation and the total number of synthetic training images were analyzed using a custom test set to identify their influence on the AM part classification accuracy. According to the findings, the optimal workflow deploys VGG16 as the network structure, includes a physics simulation to increase the classification accuracy with fewer training images, and uses 256 synthetic images to be trained for 5 epochs. Third, the proposed workflow and its generalizability were evaluated in an industrial case study, where the workflow was used to recognize previously unknown AM parts from six different build jobs. On a variety of 215 distinct part geometries and 519 recognized parts in total, classification accuracies of 99.04% (top three) and 90.37% (top one) were achieved, supporting the industrial applicability of the workflow.

The achieved results allow for the reproducibility of the workflow when implementing it for industrial applications of AM parts. While the top-one accuracy of 90.37% is not sufficient yet for full automation of AM post-processing, it is an important step toward this goal. Furthermore, the findings of this paper are relevant to similar industrial applications of visual object recognition where real training data does not exist or require high effort and expertise to be created. The only requirement is the access to the CAD data of the objects that are to be recognized, whereas the objects themselves do not have to exist yet. Furthermore, the documented findings allow for addressing specific challenges in the automated visual recognition of AM parts: mirrored and scaled parts.

These challenges will be addressed in future research by adapting training parameters and including further information in the classification process. Additionally, it is planned to increase the amount of evaluation data and extend the proposed workflow to additional AM processes than MJF. Finally, methods to recognize multiple parts in a single classification step will be explored, as this could further reduce recognition times.

**Author Contributions:** Conceptualization, J.C. and D.O.; methodology, J.C.; software, S.R.; validation, S.R., J.C. and D.O.; formal analysis, J.F.; investigation, J.C.; resources, M.M.; data curation, S.R.; writing—original draft preparation, J.C.; writing—review and editing, J.F.; visualization, J.C.; supervision, M.M.; project administration, J.C.; funding acquisition, J.C. All authors have read and agreed to the published version of the manuscript.

**Funding:** This research was funded by the Swiss Innovation Agency, Innosuisse, under grant 50383.1 IP-ENG.

**Institutional Review Board Statement:** Not applicable.

**Informed Consent Statement:** Not applicable.

**Data Availability Statement:** The test dataset is available under the following link: https://data.mendeley.com/datasets/trd8nry345/1 (accessed on 23 October 2023).

**Acknowledgments:** The authors would like to thank Wyss Zurich and Manuel Biedermann for their support of the project.

**Conflicts of Interest:** The authors Jonas Conrad, Daniel Omidvarkarjan, and Julian Ferchow were employed by the company inspire AG. The remaining authors declare that the research was conducted in the absence of any commercial or financial relationships that could be construed as a potential conflict of interest.

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
