# Peer review of "Recognition of Additive Manufacturing Parts Based on Neural Networks and Synthetic Training Data: A Generalized End-to-End Workflow"

_applsci, doi:10.3390/app132212316_

Round 1

Reviewer 1 Report

Comments and Suggestions for Authors

The manuscript proposes a generalized end-to-end workflow for automated visual real-time recognition of AM parts. Synthetic images generated from digital AM part models are used to train a neural network for image classification. The proposed workflow was evaluated in an industrial case study achieving very high part classification accuracy.

The work is very interesting and well-structured. It should be considered for publication in this journal after a minor review taking into account the following points:

·         The manuscript is well written, however there are a few acronyms that are not explained (e.g. MJF). Please review the document.

·         Explain why a rendered image resolution of 256 × 256 pixels was chosen. Increasing the resolution of images can improve the neural network to detect parts containing small details.

·         It is recommended that images with higher resolution be included, as some are difficult to read (see Figure 3).

·         Figure 10 is difficult to understand because different numbers of "images per part" were chosen for each Drops. It is recommended to modify the graph by using the same number of images for the 3 drops or create clusters for "images per part" (e.g. 32-36 or 392-400). In addition, it is recommended to introduce a line plot between the mean values to facilitate determining the trends of the graphs.

·         It is suggested to include a figure to help understand how the implementation of physics can reduce the number of training images.

·         Provide more details on the computational and processing time demand due to the implementation of the physical simulation. Is this compliant with the idea of the system being simple and fast, for industrial implementation?

Reviewer 2 Report

Comments and Suggestions for Authors

The manuscript " Recognition of additive manufacturing parts based on neural networks and synthetic training data: A generalized end-to-end  workflow" introduces an automated solution for recognizing Additive Manufacturing (AM) parts, addressing a manual bottleneck in the AM process. The authors use synthetic training images to train a neural network for image classification, enabling the recognition of AM parts without design adaptations. The workflow is generalized for different production batches and optimized for low-cost industrial applicability.

Overall, the manuscript is commendable for addressing a significant challenge in the field of additive manufacturing and will be of great interest to the readers of Applied Sciences. It should be published after addressing the minor comments below. Incorporating these suggestions and addressing the critical comments can further enhance the paper's clarity, transparency, and depth.

Minor issues: The reviewer has the following minor concerns regarding this study:

1.      The authors acknowledge certain limitations in the recognition process, such as challenges with mirrored parts and distinguishing parts that only differ in scale. It would be valuable to provide additional insights into potential strategies to address these challenges or outline future research directions aimed at enhancing classification accuracy in these cases.

2.     The manuscript would benefit from a more thorough discussion and comparison with existing literature on automated visual recognition in additive manufacturing. This would help provide context and emphasize the significance of the proposed approach in relation to prior research efforts.

3.     The manuscript could be strengthened by incorporating a more substantial concluding section. This section should succinctly summarize the key findings, discuss the broader implications of the research, and suggest future research directions. A comprehensive conclusion would help readers understand the significance and potential impact of the study.

Comments on the Quality of English Language

none

Reviewer 3 Report

Comments and Suggestions for Authors

The article is quite interesting for reader of both mechanical and computer science too.

1. Author need to take care of diagram formatting

2. Diagram quality is also poor

3. Diagram caption can be reduced to a single line of text in case of diagram 1.

4. its better to include details of the dataset was used for this research work and their availability for others.

5.  Author should provide architectural details of differ machine learning techniques like "3.2.1 Neural network structure"
